# Detection of Resistant and Enterotoxigenic Strains of *Staphylococcus warneri* Isolated from Food of Animal Origin

**DOI:** 10.3390/foods11101496

**Published:** 2022-05-20

**Authors:** Ivana Regecová, Jana Výrostková, František Zigo, Gabika Gregová, Monika Pipová, Pavlina Jevinová, Jana Becová

**Affiliations:** 1Department of Food Hygiene Technology and Safety, University of Veterinary Medicine and Pharmacy in Košice, Komenského 73, 041 81 Košice, Slovakia; ivana.regecova@uvlf.sk (I.R.); monika.pipova@uvlf.sk (M.P.); pavlina.jevinova@uvlf.sk (P.J.); Jana.Becova@uvlf.student.sk (J.B.); 2Department of Nutrition and Animal Husbandry, University of Veterinary Medicine and Pharmacy in Košice, Komenského 73, 041 81 Košice, Slovakia; frantisek.zigo@uvlf.sk; 3Department of Public Veterinary Medicine and Animal Welfare, University of Veterinary Medicine and Pharmacy in Košice, Komenského 73, 041 81 Košice, Slovakia; gabriela.gregova@uvlf.sk

**Keywords:** antimicrobial resistance, enterotoxins, food, *mecA*, *Staphylococcus warneri*

## Abstract

The topic of this work is the detection of antimicrobial resistance to *Staphylococcus warneri* strains and the genes encoding staphylococcal enterotoxins. It is considered a potential pathogen that can cause various—mostly inflammatory—diseases in immunosuppressed patients. The experimental part of the paper deals with the isolation of individual isolates from meat samples of *Oryctolagus cuniculus*, *Oncorhynchus mykiss*, *Scomber scombrus*, chicken thigh, beef thigh muscle, pork thigh muscle, and bryndza cheese. In total, 45 isolates were obtained and subjected to phenotypic (plasma coagulase activity, nuclease, pigment, hemolysis, lecithinase, and lipase production) and genotypic analyses to confirm the presence of the *S. warneri* species. The presence of genes encoding staphylococcal enterotoxins A (three isolates) and D (six isolates) was determined by PCR. Using the Miditech system, the minimum inhibitory concentration for various antibiotics or antibiotics combinations was determined, namely for ampicillin; ampicillin + sulbactam; oxacillin; cefoxitin; piperacillin + tazobactam; erythromycin; clindamycin; linezolid; rifampicin; gentamicin; teicoplanin; vancomycin; trimethoprim; chloramphenicol; tigecycline; moxifloxacin; ciprofloxacin; tetracycline; trimethoprim + sulfonamide; and nitrofurantoin. Resistance to ciprofloxacin and tetracycline was most common (73%). At the same time, out of a total of 45 isolates, 22% of the isolates were confirmed as multi-resistant. Isolates that showed phenotypic resistance to *β*-lactam antibiotics were subjected to *mecA* gene detection by PCR.

## 1. Introduction

*Staphylococcus warneri* is a coagulase-negative staphylococcal (CNS) commonly present in the microbiota of human epithelium and mucous membranes. In the last two decades, *S. warneri*, like other species of the CNS, has been reported as a new emerging pathogen that can cause serious infections, usually in connection with the presence of implanted materials but sometimes even in the absence of a foreign body and in patients considered immunocompetent. At present, there is still a lack of scientific data on the pathogenesis and epidemiology of this species [1]. According to Bhardwaj et al. [2], *Staphylococcus warneri* is a common saprophyte on human skin, present in approximately 50% of the healthy adult population.

*S. warneri* creates a biofilm on the surface of various materials (e.g., air, walls, floors, and medical equipment). In addition, it possesses various virulence factors, such as adhesion to polymeric surfaces and metabolic changes in various situations. The ability to form biofilms in foreign bodies and medical devices allows this bacterium to be resistant to antimicrobials used to treat infections [3].

The CNS, including *S. warneri*, were previously considered food-related saprophytes [4,5]. *S. warneri,* as well as other CNSs, may possess or acquire mobile pathogenic factors (e.g., exotoxins, enterotoxins, adherence factors, leukocidins, and antibiotic resistance) in the form of transposons, pathogenicity islands (PIs), plasmids, and phages similar to those of *Staphylococcus aureus*, which may lead to a potential health risk for the consumer [5,6]. The most common contamination of food with this type of staphylococcus is secondary and comes from processing at food companies as well as from the poor handling of food during the production process [5,6,7]. Most recently, *S. warneri* was detected in chicken meat, ready-for-consumption fish dishes, and raw dairy products [8,9,10]. This species is also reported as a potential source of food enterotoxicosis [11,12].

The antibiotic resistance of microorganisms is a major health, social, and economic problem for all of us today. As consumers of animal products, humans should also be aware of the risks associated with consuming products that contain bacteria resistant to some antibiotics (chloramphenicol, streptomycin, enrofloxacin). When consuming such products, there is the possibility of transferring resistance genes to the microbial population in the human intestinal tract and possible future complications in the treatment of bacterial diseases, not to mention the spread of resistance genes further in the human population and environment [13].

Antimicrobial resistance in bacteria is not a new phenomenon. Estimates for the emergence of biosynthetic pathways have been published by Wright, showing that such resistance began hundreds of millions of years ago [14]. It has also been documented that bacteria share a pool of antimicrobial-resistant genes (AMRG) conferring antibiotic resistance before human discovery and the widespread use of antibiotics. In the permafrost of 30,000 years ago, genes encoding *β*-lactam resistance and resistance to other antibiotics were documented by D’Costa [15]. There are self-transferring plasmids carrying antibiotic resistance genes (ARGs) that can infect many phylogenetically distinct bacteria, forming a group of genes that can be shared [16]. Methicillin-resistant coagulase-negative staphylococci (MR-CoNS) are a major cause of infectious diseases [3].

The CNS, including *S. warneri*, are usually resistant to multiple drugs, have limited treatment options, cause incurable diseases, and become reservoirs of resistant strains in hospitals. The cause of antimicrobial resistance is multi-factorial, ranging from a lack of infection prevention to the irrational use of antimicrobials by health-care professionals and patients. Recent studies suggest that CNSs were highly resistant to penicillin, methicillin, vancomycin, oxacillin, and erythromycin. CNSs were less resistant to cephalosporins, aminoglycosides, and quinolones [17].

A significant proportion of *S. warneri* isolates belonging to the CNS group have a high degree of genetic diversity, indicating a high predisposition to antimicrobial resistance [18]. Biofilm isolates can carry multiple genes encoding resistance to beta-lactams, aminoglycosides, and macrolide–lincosamide antibiotics. Antibiotic resistance genes were found to be more common in biofilm-positive than biofilm-negative isolates of *S. warneri*. Biofilm structure, due to cell aggregation, may be ideal for horizontal gene transfer and thus facilitate the spread of antibiotic resistance [18].

A study by Chaves et al. [19] shows that CNS are capable of producing enterotoxins, confirming data previously published by other authors. Although current legislation only recommends counting coagulase-positive staphylococci, the study points to the possibility of reformulating existing microbiological standards. In addition, a new approach that correlates the levels and intervals of enterotoxin production could be used for a new and safer food policy.

The aim of our study was to point out the presence of resistant and enterotoxigenic isolates of *S. warneri* from various food commodities of animal origin. In particular, the study focused on the presence of genes encoding for the production of staphylococcal enterotoxins A–E. At the same time, the aim was to determine the MIC for different groups of antibiotics used in human and veterinary medicine and subsequently to detect methicillin-resistant staphylococci carrying the *mecA* gene.

## 2. Materials and Methods

Food samples, 4 from each species, investigated in this study derived from the following areas: Atlantic mackerel muscle (*Scomber scombrus*) originating in FAO 27 (Ireland), rainbow trout muscle (*Oncorhynchus mykiss*) originating in the South Bohemian Region (Czech Republic), wild rabbit muscle (*Oryctolagus cuniculus*) originating from a joint hunt on the grounds of the University facility for breeding and diseases of game, fish, and bees in Rozhanovce (Košice, Slovakia), samples of beef thigh muscle, pork thigh muscle (*Sus scrofa domesticus*) and thigh muscles from chickens (*Gallus gallus domesticus*) were provided from slaughterhouses located in eastern Slovakia. The samples of full-fat bryndza cheese were provided from two dairy farms located in central Slovakia.


**Staphylococcus identification**


### 2.1. Detection of the Phenotypic Properties of the Isolates of Isolation of Strains


Free coagulase detection


Staphylococcal colonies were inoculated from the blood–agar surface with sterile bacterial ashes into test tubes with 10 mL of broth containing brain–heart infusion (BHI broth; OXOID, Hampsire, UK). After incubation for 18–24 h at 37 °C, 0.1 mL of multiplied broth culture of the tested strains was added to test tubes with 1 mL of sterile rabbit plasma (StaphyloPK test, IMUNA, Šarišské Michaľany, Slovakia). The inoculated plasma was incubated at 37 °C. The results were read after 1, 2, 3, 6, and 24 h to capture the delayed positive reaction that may occur within 24 h.


Deoxyribonuclease production


DNase agar (OXOID, Hampshire, UK) containing DNA was used to detect nuclease activity. Isolates with positive nuclease activity hydrolyzed the DNA contained in the agar after 24 h incubation at 37 °C. Positive nuclease activity manifested itself on the surface of DNase agar after pouring the surface of the medium with 1 N hydrochloric acid solution as a transillumination zone. If no DNA hydrolysis occurred near the inoculated strains, the soil became cloudy due to HCl.


Other phenotypic properties of isolates


Pigment production as well as hemolysis of individual isolates were evaluated after 24 h of incubation on the agar surface with a 2% addition of lamb blood. Baird–Parker selective diagnostic medium was used to detect the formation of lecithinase and lipase on the surface of which the identified staphylococcal strains were inoculated with sterile bacterial ashes. The inoculated petri dishes were incubated at 37 °C/24 h. After incubation, in the positive case, a precipitation zone around the inoculated colonies in the agar medium in the presence of lecithinase can be detected. This is due to the hydrolysis of the lecithin phospholipid to 1.2-diglyceride and phosphorylcholine [20].

In the presence of the enzyme lipase, a brightening zone was created around the colonies. This is due to the cleavage of triacylglycerides to triacylglycerol and fatty acids, which are present in Baird–Parker agar medium.

### 2.2. Isolation DNA from Staphylococcal Isolates

Total genomic DNA was isolated according to Sharma et al. [21] from staphylococcal isolates grown in BHI broth. The pellet obtained after centrifugation of 1.5 mL of expanded staphylococcal culture (12,500× *g*/5 min) was resuspended in 100 μL of 0.5% Triton X-100 (Koch-Light Lab., Suffolk, VA, USA) and centrifuged again (12,500× *g*/5 min). Subsequently, the sediment was resuspended in 200 µL of 0.5% Triton X-100. The samples thus prepared were incubated at 95 °C/10 min and centrifuged again (12,500× *g*/5 min). The obtained supernatant was used as a source of DNA in PCR reactions.

### 2.3. Genotypic Analysis of Staphylococcal Isolates

The identification of individual staphylococcal strains was performed using the PCR method. The identification of individual staphylococcal strains was performed by using specific primers the SwarF (TGTAGCTAACTTAGATAGTGTTCCTTCT) and SwarR (CCGCCACCGTTATTTCTT) synthesized by Amplia s.r.o. (Bratislava, Slovakia) according to Iwas et al. [22]. Commercially produced Hot FIREPol^®^ MasterMix (Amplia s.r.o., Bratislava, Slovakia) was used in PCR reactions. The total reaction mixture volume of 20 µL contained 5 ng/µL of template DNA and each primer at a concentration of 10 pmol. The finished mixture was heated at 95 °C for 12 min at initial denaturation, then heated over 30 amplification cycles: 95 °C/30 s, annealing 60 °C/30 s and 72 °C/2 min, using a thermal cycler (TECHNE TC-512, London, UK) with an extension of 10 min/72 °C. The 65 bp PCR product, which was amplified, was collected for 5 μL for analysis on a 1.5% agarose gel in TBE buffer (Tris-borate-EDTA). GelRed TM (Biotium, Fremont, CA, USA) was added to the agarose gel to visualize the PCR fragments using a Mini Bis Pro^®^ reader (DNR Bio-Imaging Systems Ltd., Neve Yamin, Israel). The obtained PCR products were sequenced at the European Custom Sequencing Center GATCBiotech AG (Cologne, Germany) and compared with the nucleotide sequences of the reference strains in GenBank NCBI. The reference strain *S. warneri* CCM 2730T (Czech Collection of Microorganisms, Brno, Czech Republic) was used as a positive control for PCR strain identification.

### 2.4. Detection of Genes Encoding Staphylococcal Enterotoxins

To detect genes encoding staphylococcal enterotoxin (SE) production (*sea, seb, sec, sed,* and *see*), a multiplex PCR assay, according to Sharma et al. [21] was performed by using universal forward primer and five specific reverse primers for each staphylococcal enterotoxin to detect each of the five enterotoxins.

The reaction mixture used and the reaction conditions were the same as described above (except for the annealing temperature, which was 45 °C). The sizes of the PCR products are shown in Table 1.

The following reference strains of *Staphylococcus aureus* (CCM, Brno, Czech Republic) were used as a positive control for this multiplex PCR: CCM 5756 (*sea* gene), CCM 5757 (*seb* gene), CCM 5984 (*sec* gene), CCM 5973 (*sed* gene), and CCM 5972 (*see* gene).

### 2.5. Detection of Antimicrobial Resistance

Eighteen isolates of *S. warneri* were analyzed for their antibiotic susceptibility. Minimal inhibitory concentrations (MIC) were determined according to CLSI document [23] and EUCAST document [24], using a Miditech system (Bratislava, Slovakia) with interpretive reading of MIC. The antibiotics used in this study were as follows: ampicillin + sulbactam (SAM), piperacillin + tazobactam (TZP), oxacillin (OXA), erythromycin (ERY), clindamycin (CLI), teicoplanin (TEC), vancomycin (VAN), rifampicin (RIF), gentamicin (GEN), linezolid (LNZ), ciprofloxacin (CIP), moxifloxacin (MFX), tetracycline (TET), tigecycline (TGC), chloramphenenicol (CHL), trimethoprim (TMP), trimethoprim + sulfonamide (COT), and nitrofurantoin (NIT). 

In isolates showing phenotypic resistance to oxacillin, the presence of the *mecA* gene was detected by PCR according to Poulsen et al. [25]. The primers used in this PCR reaction were MecA1 (GGGATCATAGCGTCATTATTC) and MecA2 (AACGATTGTGACACGATAGCC) (Amplia sro, Bratislava, Slovakia). The PCR reaction conditions were the same as for the species identification of staphylococci mentioned above. The resulting size of the PCR product was 527 bp. The reference isolate *S. aureus* CCM 4750 (Czech Collection of Microorganisms, Brno, Czech Republic) was used as a reference strain for the PCR and for detection of MIC in this study.

## 3. Results

Typical and atypical colonies were harvested from the surface of the Baird–Parker agar medium after incubation for the initial culture screening of individual samples. A total of 560 staphylococcal isolates were obtained and subjected to further identification. All isolates obtained were subjected to detection of coagulase activity. In the detection of this phenotypic trait, for 200 isolates the result of coagulase assay was negative, showing no coagulase activity. Therefore, these isolates were classified as CNS. Subsequently, in this group of isolates, the detection of other phenotypic characteristics was carried out for a better characterization of the *S. warneri* strains as well as for detection of virulent strains: nuclease activity, pigment, hemolysis, lecithinase, and lipase production (Table 2). In isolates included in the CNS, genotypic identification was performed using the PCR method, where 45 isolates were identified (22% of the CNS group) as *S. warneri*. It was retrospectively evaluated that 8 isolates derived from brown trout, 4 isolates from Atlantic mackerel, 7 isolates from pork thigh muscle, 2 isolates from beef thigh muscle, 14 isolates of *S. warneri* came from wild rabbit, 7 isolates from chicken thigh muscle and, 3 isolates came from bryndza cheese.

As shown in Table 2, gray pigment was observed most frequently on blood agar. The production of a yellowish-white coloration of *S. warneri* colonies also appeared. The production of α-hemolysis was also confirmed in these isolates. *β*-hemolysis was not confirmed in any of the isolates. Similarly, nuclease production on DNAse agar was not confirmed in any isolate. However, lipase production was confirmed in a large number of *S. warneri* isolates. Lipase production was manifested by a transillumination zone in Baird–Parker agar medium around the overgrown colony. In contrast, a smaller number of isolates produced the enzyme lecithinase. Its production appeared as a precipitating ring around the overgrown colony on Baird–Parker agar medium. The detection of lecithinase and lipase production confirmed their current production in three isolates of *Oncorhynchus mykiss*, two isolates from *Oryctolagus cuniculus*, and one isolate each from *Scomber scombrus*, pork thigh muscle, and Bryndza cheese.

Subsequently, after the detection of individual phenotypic manifestations, genes encoding the staphylococcal enterotoxins SEA, SEB, SEC, SED, and SEE were detected. As shown in Figure 1, *S. warneri* isolates confirmed the presence of genes encoding the production of SEA and SED enterotoxins. The presence of the *sea* gene was confirmed in three isolates and the *sed* gene in six isolates. Of these, the co-presence of the *sea* and *sed* genes was detected in two isolates. It was an isolate isolated from a sample of *Oryctolagus cuniculus* and a sample of Bryndza cheese. At the same time, these two isolates phenotypically showed α-hemolysis and lecithinase and lipase production.

As seen in Figure 2A, we were unable to detect the presence of genes encoding enterotoxin production in *S. warneri* isolates by multiplex PCR, only in the reference strains. For this reason, we approached the detection of individual genes separately using primers from multiplex PCR. As seen in Figure 2B,C, the presence of the *sea* and *sed* genes was subsequently confirmed.

After detection of phenotypic characteristics and detection of toxinogenic isolates, MIC detection of antibiotics was performed using the Miditech system. In general, as shown in Table 3, the highest resistance to ciprofloxacin and tetracycline was confirmed. The MIC50 of ciprofloxacin was 2.0 mg/L and the MIC90 was 4.0 mg/L while the MIC of GX was 1.2 mg/L. For tetracycline, the MIC50 was 16.0 mg/L and the MIC90 was 32.0 mg/L. At the same time, the MIC GX of this antibiotic was 8.3 mg/L (Figure 3). At the same time, intermediate sensitivity was confirmed in all 45 isolates against trimethoprim + sulfonamide, and nitrofurantoin. The strains on vancomycin, trimethoprim, trimethoprim + sulfonamide, and nitrofurantoin appeared to be the most sensitive overall (Table 3).

Specifically, isolates isolated from the *Oncorhynchus mykiss* muscle (eight isolates) were the most frequently confirmed to be resistant to tetracycline (87.50%). Resistance to clindamycin and gentamicin (37.50%) was also among the common resistance detected in these isolates. Isolates isolated from *Scomber scombrus* muscle samples (four isolates) also showed the most common resistance to tetracycline (75.00%) and to gentamicin, ciprofloxacin, and chloramphenicol (50.00%). *S. warneri* strains isolated from the pork thigh muscle (7 isolates) and the *Oryctolagus cuniculus* muscle (14 isolates) showed the most common resistance to tertracycline (100.00%/92.86%). Isolates isolated from chicken thigh muscle (seven isolates) showed the highest resistance to ciprofloxacin (85.71%). In isolates derived from beef thigh muscle (two isolates), 100.00% resistance to clindamycin, rifampicin, and gentamicin was observed. The resistance of all *S. warneri* isolates (three isolates) to rifampicin (100.00%) was confirmed in the last examined bryndza samples.

Based on the antimicrobial resistance using the Miditech system, the various resistance mechanisms, which are shown in Table 4, were confirmed. As can be seen from the table, incomplete fluoroquinolone resistance was generally the most common. Specifically, this resistance mechanism was most commonly detected in isolates isolated from *Oryctolagus cuniculus* samples (13 isolates/92.86%). The presence of methicillin-resistant coagulase-negative staphylococci (MRCNS) in 13.33% of the examined isolates was also phenotypically confirmed.

Based on the phenotypic expression of these isolates (six isolates), the *mecA* gene, which encodes one of the most common mechanisms of methicillin resistance, was detected. Based on the results of the PCR reaction and subsequent sequencing of the PCR products, the presence of the *mecA* gene was confirmed in four isolates (66.66%). These isolates were isolated from samples of *Scomber scombrus*, *Oryctolagus cuniculus*, thigh muscle of chickens, and Bryndza cheese. The isolates also exhibited virulence phenotypic properties such as α-hemolysis, lipase, and lecithinase production. All isolates carrying the *mecA* gene produced a yellow-white pigment. At the same time, the presence of the *sea* gene (1 isolate), the *sed* gene (1 isolate), and both genes (2 isolates) was confirmed in these isolates.

Based on the detection of resistance of *S. warneri* isolates, multi-drug resistance was also confirmed in 10 isolates (Table 4). Specifically, two isolates isolated from *Oncorhynchus mykiss* samples were confirmed to be resistant to four antimicrobial groups at the same time, namely RIF-GEN-CIP-TET. In isolates derived from *Oryctolagus cuniculus* samples, multi-drug resistance was confirmed in two isolates, namely GEN-CIP-TET. The other two isolates confirmed simultaneous resistance to CIP-TET-CHL-RIF-GEN, of which one isolate also carried the *mecA* gene. In isolates derived from pork thigh muscle, the co-resistance to GEN-CIP-TET was confirmed in three isolates. Multi-resistance was also confirmed in one isolate isolated from Bryndza cheese samples against OXA-ERY-CIP-CHL.

## 4. Discussion

The isolation of food samples produced 45 isolates that were subsequently tested for phenotypic and genotypic characteristics to confirm that they were *Staphylococcus warneri,* being catalase-positive, oxidase-negative, and coagulase-negative. This species is also a commensal bacterium that occurs as part of the skin microbiota in humans and animals. Although *S. warneri* accounts for less than 1% of the total staphylococcal population, it is responsible for a variety of human diseases, such as immune suppression, skin, eye, and urinary tract infections, nosocomial infections, and weakening the immune systems of patients and neonates [1]. However, in a study by Hoveida et al. [26], *S. warneri* accounted for up to 7.5%, similar to our study, where the incidence rate was up to 22% of the total identified CNS population. In our study, such a higher incidence of *S. warneri* may be due to a different origin of the samples. The samples came from bryndza, game, fish, and thigh muscles of livestock. The microbiota of these commodities can be affected by hygiene levels, where the role of food processors is crucial in determining the hygienic status of the final product, with mishandling leading to an increased likelihood of microbial contamination by humans, including multi-drug-resistant and/or enterotoxigenic staphylococci [27].

In our study, phenotypic and genotypic identification was used to identify this species. In terms of phenotypic characteristics, hemolytic activity was tested, which was demonstrated in 10 isolates in the form of *α*-hemolysis. This confirms the production of cytolytic alpha-toxin, which is also known as alpha-hemolysin. By binding to the cell surface, it causes necrotic cell death [28]. Alpha-hemolysin production was also confirmed in a study by Noumi et al. [29], in two tested *S. warneri* isolates.

Pigment formation was demonstrated in 36 isolates, where the most frequently produced was a gray-white pigment, to a lesser extent a yellow-white pigment. According to Becker et al. [4], most CNS species are usually unpigmented. Pigmented colonies are characteristic only for the species *S. chromogenes, S. devriesei, S. lugdunensis, S. sciuri, S. vitulinus, S. warneri,* and *S. xylosus*. They form a pigment of gray-white, gray-yellow, yellow, or yellow-orange color.

Other monitored properties were the production of hydrolytic enzymes (lecithinase and lipase). Hydrolytic enzymes are among the CNS virulence factors that contribute to soft-tissue degradation and aid in biofilm formation [30]. Lecithinase was observed in our study in 17 *S. warneri* isolates. It is a type of phospholipase that acts on lecithin [30]. Lipase production was confirmed in 35 *S. warneri* isolates. Staphylococcal lipases are usually released outside the cell, where they perform several functions, some of which are of hygienic importance. Staphylococcal extracellular lipases are overexpressed in connection with pathogenic events. They are also recognized as contributors to human and animal pathogens by improving bacterial nutrition on the skin, disrupting host cell membranes, disrupting the immune response, and host cell signaling [31]. Many studies have been conducted to examine staphylococcal lipase production, and these bacteria have been found to have high levels of lipase activity in meat and dairy production [32,33], as confirmed by our study. Lipolytic bacteria are classified into different genera based on their gene sequences and biochemical properties, and high lipase levels are also a common feature of staphylococci, including *S. warneri* [34]. The presence of the abovementioned hydrolytic properties was confirmed in *S. warneri* isolates by Noumi et al. [29].

Subsequently, the PCR method, which is currently considered to be the most accurate identification method, was used to accurately identify the *S. warneri* isolates obtained. The *16S rRNA* gene is generally used as a target sequence to identify species in the genus Staphylococcus. However, this *16S rRNA* gene in *Staphylococcus epidermidis* is very similar to the sequence of other *16S rRNAs* in other CNS species. To solve this problem, it is possible to use an alternative target sequence that shows a greater sequence divergence than the *16S rRNA* gene. Recently, a highly conserved, ubiquitous *sodA* gene was used that encodes manganese-dependent superoxide dismutase [35] In our study, 45 *S. warneri* isolates from 200 identified CNSs were confirmed using a specific *sodA* gene sequence. The presence of genes encoding the production of staphylococcal enterotoxins A to E was subsequently detected in the *S. warneri* isolates.

SE production is one of the most notable virulence factors in staphylococci. Staphylococcal enterotoxins (SE) are mainly divided into five classical serological types: SEA, SEB, SEC, SED, and SEE as well as the other recently discovered SEG, SHE, SEI, SER, SES, SET, and the enterotoxin-like proteins, such as SElK, SElN, SE10, SE1P, SE1Q, and SElU [36,37]. Reliable detection of SE genes has a dual function. First, it helps in the genotyping of coagulase-positive staphylococci (CPS) for epidemiological studies. Second, it provides an assessment of the possible occurrence of SE genes in CNS strains which pose a potential risk of staphylococcal enterotoxicosis to consumers [36]. In our study, the presence of the *sea* (6% of isolates) and *sed* (13% of isolates) genes was confirmed. De Freitas Guimarães et al. [38] also confirmed the presence of *sea* and *sed* genes in *S. warneri* isolates isolated from food of animal origin but in higher percentages than those confirmed in our study. In further studies, they confirmed the presence of other genes that encode enterotoxins, namely *sec* and *she* [29,39]. 

Banaszkiewicz et al. [39] and Hu et al. [40] also suggest the detection of the continuous transfer of elements containing SE genes from staphylococci which contain the stable enterotoxin genes of the CNS originating from wild animals, which was confirmed by our study.

Staphylococcal isolates of phenotypically and genotypically identified isolates were subsequently determined using the Miditech system, where the highest resistance was recorded against CIP and TET. Similarly, frequent resistance to TET was confirmed in the Hoveida et al. [41] study, which accounted for up to 50% of the resistance.

At the same time, our study confirmed the presence of *S. warneri* MRS in 13.33% of the *S. warneri* isolates. On the basis of the detection of the MRSCNS phenotypic expression, the *mecA* gene was detected in these isolates. Methicillin resistance is associated with the presence of the *mecA* gene, which encodes additional binding to a penicillin protein (PBP2A or PBP2’). This protein has a lower affinity for all beta-lactam antibiotics. The *mecA* gene is found on a mobile genetic element called the staphylococcal cassette chromosome mec (SCCmec) [42]. Another mechanism in staphylococcal resistance to beta-lactams is beta-lactamase production, encoded by the *blaZ* gene [43]. In our study, the presence of the *mecA* gene was confirmed in 66.66% of isolates (four isolates) that phenotypically appeared as MRSCNS (six isolates). Humphries et al. [44] confirmed the presence of the *mecA* gene in *S. warneri* isolates in a similarly high percentage. The *mecA*-positive *S. warneri* isolates accounted for up to 41.66% of the total of 48 isolates tested. Similarly, Hoveida et al. [41] confirmed the prevalence of *mecA* genes in *S. warneri* in 30% of the 40 isolates isolated from different types of food.

In addition to the increasing incidence of MRSCNS isolates, resistance to aminoglycosides has recently increased. Aminoglycoside resistance has increased, especially among methicillin-resistant strains carrying the *mecA* gene [45]. Inactivation of antibiotics by aminoglycoside-modifying enzymes (AME), which are encoded by genetic elements, is a major pathway for resistance [46]. The most important of these enzymes is aac (6′)/aph (2″), which alters aminoglycosides of medical importance, such as tobramycin and gentamicin [47]. This type of resistance to aminoglycosides was also confirmed in our investigated CNS isolates, specifically in 33.33% of the *S. warneri* isolates. Of these, MRSCNS was identified in three strains; they showed resistance to aminoglycosides using combined enzymatic resistance to tobramycin, gentamicin, and amikacin. Consistent with many other studies [47,48,49], the enzyme aac (6′)/aph (2″) is the most common, where 85% of AME-positive isolates were found. 

Another resistance detected in the isolates we examined was constitutive resistance to macrolides, lincosamides and streptogramin B (cMLSB). However, studies mostly focus on the most common isolated staphylococci, i.e., *S. aureus* and *S. epidermidis* [50,51,52]. Recently, other staphylococcal species, such as *S. hominis*, *S. haemolyticus*, *S. warneri* and *S. simulans,* have emerged as etiological factors in serious human infections. The phenotypic expression of resistance to MLSB in these staphylococci may be inducible and manifest in clinical resistance to lincosamides and streptogramin B induced by 14- and 15-membered macrolides or constitutive, determining resistance to all MLSB antibiotics [53,54]. In our study, only constitutive macrolide resistance was phenotypically confirmed (2.22%). Similarly, a low percentage (10%) were confirmed cMLSB in *S. warneri* in a study by Szemraj et al. [55].

Incomplete fluoroquinolone resistance was also confirmed in the *S. warneri* isolates in our study, with up to 71.11%; in other words, this is a mutation with incomplete resistance to fluoroquinolones. However, little is known about the mechanisms involved in the development of fluoroquinolone resistance (through exogenous acquisition, de novo mutation, or selection of a minority-resistant mutant) in CNS [56]. However, previous studies have confirmed that methicillin-resistant strains developed resistance to fluoroquinolones more rapidly in *S. aureus* and the CNS than in methicillin-sensitive strains. This difference is partly explained by nosocomial transmission in some environments and thus the potential for co-selection with several antimicrobials (due to the common multi-drug resistance phenotype of MRS isolates [57]. Resistance to more than one antibiotic was also confirmed by Persson-Waller et al. [58], where multi-drug resistance occurred in 9% of the total 56 CoNS isolates. Nunes et al. [59] also confirmed 14 multi-drug-resistant CNS. Our demonstrated multi-drug resistance of CNS isolates, specifically *S. warneri*, is consistent with previous studies on coagulase-negative staphylococci, which found several resistant and multi-drug resistant staphylococcal isolates [60]. Our results are supported by the study of Senga et al. [61], who reported up to 80% resistance to several types of antibiotics in CNS isolates. 

## 5. Conclusions

The results of the paper point to the ever-increasing incidence of resistant and multi-resistant and enterotoxigenic isolates of *S. warneri* in foodstuffs of animal origin, especially game and wild fish. These results point to the risk of the presence of enterotoxigenic and resistant isolates of *S. warneri* in the food industry, as they can serve as vectors for the transfer of resistance determinants into the genomes of bacteria inhabiting the consumer’s digestive tract. Therefore, the rational use of antibiotics, preventive measures in environmental hygiene, and the monitoring of antibiotic resistance are important prevention measures to prevent the spread of antimicrobial resistance. At the same time, the presence of enterotoxigenic isolates indicates a potential risk for staphylococcal enterotoxicosis in humans.

## Figures and Tables

**Figure 1 foods-11-01496-f001:**
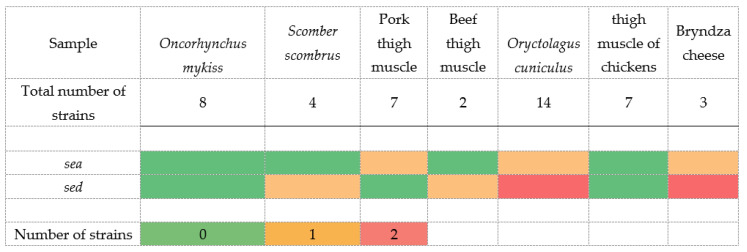
Heatmap demonstration of the presence and absence of genes encoding staphylococcal enterotoxins.

**Figure 2 foods-11-01496-f002:**
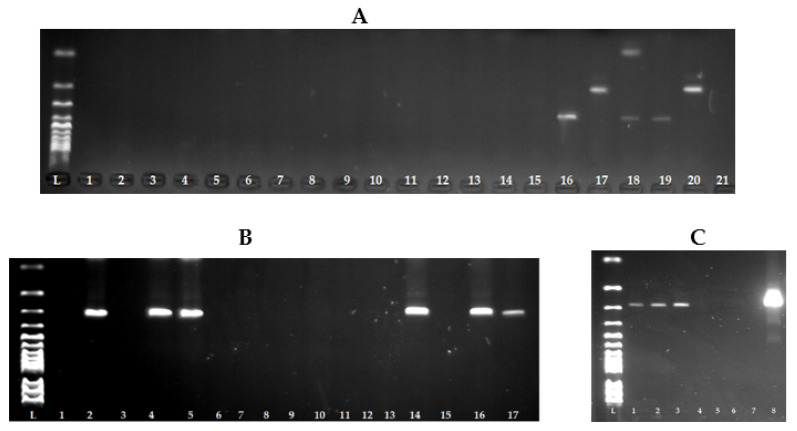
Detection of genes encoding the staphylococcal enterotoxins of *Staphyoccocus warneri* using the PCR method. (**A**) L: 100 bp ladder; lines 1–15: isolates *S. warneri* without genes encoding the staphylococcal enterotoxins; line 16: reference strain for *sea* gene CCM 5756 *S. aureus* (270 bp); line 17: reference strain for *seb* gene CCM 5757 *S. aureus* (165 bp); line 18: reference strain for *sec* gene CCM 5984 *S. aureus* (102 bp); line 19: reference strain for *sed* gene CCM 5973 *S. aureus* (306 bp); line 20: reference strain for *see* gene CCM 5972 *S. aureus* (213 bp); line 21: negative control. (**B**) L: 100 bp ladder; line 1: negative control; line 2: reference strain for *sed* gene CCM 5973 *S. aureus* (306 bp); lines 4, 5, 14, 16, 17: isolates *S. warneri* with *sed* gene (306 bp). (**C**) L: 100 bp ladder; lines 1–3: isolates *S. warneri* with *sea* gene (270 bp); line 7: negative control; line 8: reference strain for *sea* gene CCM 5756 *S. aureus* (270 bp).

**Figure 3 foods-11-01496-f003:**
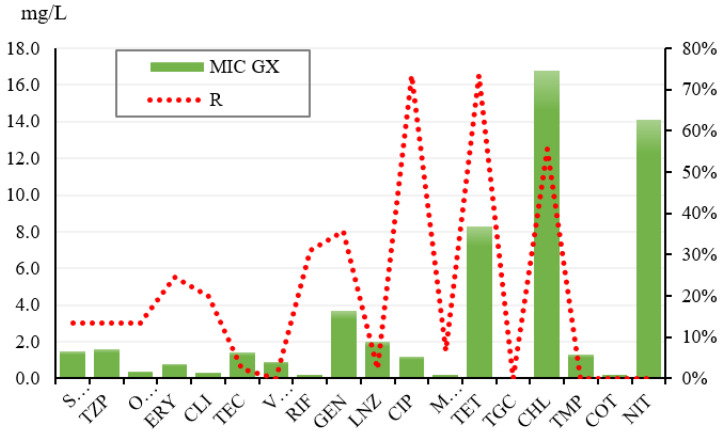
Overview of MIC and MIC GX in resistant *Staphylococcus warneri* isolates. GX MIC—Geometric average MIC—the average MIC value at which the strains are inhibited.

**Table 1 foods-11-01496-t001:** Primers used in multiplex PCR to detect staphylococcal enterotoxins [21].

Gene	Primer	Nucleotide Sequence 5′-3′	Product Size (bp)
universal	Sauni-F	TGTATGTATGGAGGTGTAAC	
*sea*	SAA -R	ATTAACCGAAGGTTCTGT	270
*seb*	SAB -R	ATAGTGACGAGTTAGGTA	165
*sec*	SAC -R	AAGTACATTTTGTAAGTTCC	102
*sed*	SAD -R	TTCGGGAAAATCACCCTTAA	306
*see*	SAE -R	GCCAAAGCTGTCTGAG	213

bp—base pairs.

**Table 2 foods-11-01496-t002:** Virulence phenotypic properties of *S. warneri* isolates.

	Number of Strains	Pigment				Hemolysis	Lecithinase	Lipase	Nuclease
		White (without Pigment)	Gray	Gray-White	Yellow-White				
** *Oncorhynchus mykiss* **	**8**	**0**	**2**	**6**	**0**	**0**	**3**	**7**	0
** *Scomber scombrus* **	4	0	2	1	1	3 (*α*)	2	3	0
**Pork thigh muscle**	7	0	0	6	1	0	3	1	0
**Beef thigh muscle**	2	1	1	0	0	0	1	5	0
** *Oryctolagus cuniculus* **	14	0	3	7	4	5 (*α*)	4	12	0
**thigh muscle of chickens**	7	1	0	4	2	1 (*α*)	1	0	0
**Bryndza cheese**	3	1	1	1	0	1 (*α*)	3	3	0
**Total**	**45**	**9**	**6**	**23**	**7**	**10**	**17**	**31**	**0**

(α)—*α*-hemolysis.

**Table 3 foods-11-01496-t003:** Percentage of antimicrobial resistance.

ATB	S	I	R	GX SI	MIC_50_	MIC_90_	Total
**SAM**	86.67%	0.00%	13.33%	0.9	1.00	64.00	45
**TZP**	86.67%	0.00%	13.33%	0.8	1.00	128.00	45
**OXA**	86.67%	0.00%	13.33%	0.2	0.25	8.00	45
**ERY**	75.56%	0.00%	24.44%	0.3	0.50	16.00	45
**CLI**	89.00%	0.00%	20.00%	0.1	0.25	8.00	45
**TEC**	97.78%	0.00%	2.22%	1.3	1.00	4.00	45
**VAN**	100.00%	0.00%	0.00%	0.9	1.00	2.00	45
**RIF**	42.22%	26.67%	31.11%	0.1	0.50	2.00	45
**GEN**	64.44%	0.00%	35.56%	0.4	0.50	256.00	45
**LNZ**	97.78%	0.00%	2.22%	2.0	2.00	4.00	45
**CIP**	0.00%	26.67%	73.33%	0.2	2.00	4.00	45
**MFX**	93.33%	0.00%	6.67%	0.1	0.13	0.25	45
**TET**	26.67%	0.00%	73.33%	0.4	16.00	32.00	45
**TGC**	100.00%	0.00%	0.00%	0.1	0.06	0.13	45
**CHL**	44.44%	0.00%	55.56%	5.3	16.00	64.00	45
**TMP**	0.00%	100.00%	0.00%	1.3	2.00	2.00	45
**COT**	100.00%	0.00%	0.00%	0.2	0.25	0.50	45
**NIT**	0.00%	100.00%	0.00%	14.1	16.00	16.00	45

S—sensitive strain, I—intermediate sensitive strain, R—resistant strain; GX SI MIC—Geometric mean MIC of strains that can be treated with a given ATB (S + I). Mean MIC of treatable strains. MIC_50_—value which expresses the minimum inhibitory concentration of a given antibiotic at which at least 50% of the population is inhibited. MIC_90_—value which expresses the minimum inhibitory concentration of a given antibiotic at which at least 90% of the population is inhibited.

**Table 4 foods-11-01496-t004:** Phenotypic detection of various mechanisms of resistance.

Mechanisms of Resistance	Number	%
MRCNS	6	13.33%
Aminogl.PH(2″)-AC(6′)	15	33.33%
Fluoroq.incompl.resistance	32	71.11%
Constitutive MLSB/c	1	2.22%
Inducible MLSB/i	0	0.00%
Multi-resistance	10	22.22%

MRCNS: methicillin-resistant coagulase-negative staphylococci; Amngl.PH (2″)-AC (6′): combined enzymatic resistance to GEN, TOB, and AMI; constitutive MLSB/c: constitutive resistance to macrolides, lincosamides, and streptogramin B; inducible MLSB/i: inducible resistance to macrolides, lincosamides, and streptogramin B; multi-resistance: current resistance in 3 or more unrelated ATB groups; fluoroq.incompl.resistanc: incomplete fluoroquinolone resistance, mutation with incomplete resistance to fluoroquinolones.

## Data Availability

The data presented in this study are available upon request from the corresponding author.

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
