# Peer review of "Detection of Resistant and Enterotoxigenic Strains of Staphylococcus warneri Isolated from Food of Animal Origin"

_foods, 2022, doi:10.3390/foods11101496_

Round 1

Reviewer 1 Report

General comments

The manuscript provides a phenotypic and genotypic characterization of S. warneri isolated from food samples in order to detect incidence of resistant and multi-resistant and enterotoxigenic strains in foodstuffs of animal origin. In general, the work is sufficiently appropriate. Except for Discussion section, English and scientific sound is often poor, and some parts are written carelessly. Names of microorganisms are written carelessly. The manuscript needs major and though linguistic corrections. Please carefully proof-read spell check the entire manuscript to eliminate grammatical errors. Check Latin words and microorganisms’ names that they must be written in italic. Please, check them throughout the entire manuscript.

The experimental part is adequately described, however I suggest to improve data presentation, especially related to the genotypic characterization, for example by adding a figure showing agarose gel of multiplex-PCR result…

I suggest to major revised the manuscript.

In the Introduction the authors referred to biofilm production by S. warneri as a virulence factor that can confer resistance, it could be interested to investigate the selected resistant and multi-resistant strains on their capacity to form biofilm.

Few suggestions/critical advice are reported below:

Line 35 change flora with microbiota or microbes associated to the human epithelium

Lines 42-43 repeated concept

Line 44 specify which type of materials or surfaces

Line 44 produce organic acids? If it is possible specify which?

Line 51 some antibiotics

Line 59 rephrased the sentence “…antimicrobial resistant genes (AMRG) conferring antibiotic resistance ….

Line 64 cancel the word Schluter

Line 67 incurable diseases

Line 74 add references

Line 78 I advise to use a more scientific language structure as “Biofilm structure, due to cell aggregation….”

Line 87 In particular, the study focused….

Line 88 encoding for the production..

Line 93 Improve english: “Food samples, 4 from each species, selected for or investigated in this study derived from the following areas…”

Lines 99-100 change “came from” with “derived from” or “were provided from”

Line 112 within 24 hours

Line 127 The inoculated petri dishes were incubated …

Line 128 …a precipitation zone around the inoculated colonies ….can be detected.

Line 137 I would remove this sentence, it is not necessary

Lines 138-140 rephrase the sentences “the identification of individual staphylococcal strains was performed by using genus-specific primers or species-specific primers ?: SwarF….and SwarR….., synthetized…..

Line 159 A multiplex PCR assay, according to….was performed by using one universal….

Line 174 “Antibiotic susceptibility” is repeated on the same sentence, please correct

Line 177 the antibiotics used in this study

Line 188 S. aureus should be written in italic

Line 194 rephrase the sentence “after incubation for the initial culture screening

Line 195 remove “in this way”

Line 197 please, rephrase the sentence “for 200 isolates the result of coagulase assay was negative, showing no coagulase activity.

Line 200 for a better characterization of S. warneri strains

Line 202 please correct the concept “genotypic identification was performed”

Line 204 “8 strains were isolated from” or “derived from”

Table 2 I advise to change the caption of table 2 in a more appropriate way by removing “Presentation of…” like “Virulence phenotypic properties of S. warneri isolates” in the table change “summary” with TOTAL

Lines 211-214 S. warneri should be written in italic

Lines 218-221 Latin words should be written in italic

Figure 1 Please, add a color legend to help the reader

Lines 227-228 please, genes should be written properly please check all genes names throughout the manuscript

Table 3 change Summary with TOTAL

Figure 2 Please translate the Fig 2 caption in English

Line 255 S. warneri strains isolated from….

Lines 250-260 check carefully Latin and microbes’ names

Table 4 check carefully table 4

Line 299 This species is ….it is also a commensal bacterium that occurs ….

Line 300 skin microbiota

Line 304 one reference is missing

Line 308 change the word “microflora” with “microbiota”

Author Response

Jana Výrostková, DVM, PhD.

Department of Food Hygiene, Technology and Safety

University of Veterinary Medicine and Pharmacy

Komenského 73

041 81 Košice

Slovak Republic

E-mail: jana.vyrostkova@uvlf.sk

April 22, 2022

Editorial Board

Foods

MDPI

Postfach, CH-4020 Basel,

Switzerland

Dear Reviewer

Please find attached our revised research article Detection of resistant and enterotoxigenic strains of Staphylococcus warneri isolated in food of animal origin “ written by Ivana Regecová, Jana Výrostková, František Zigo, Gabika Gregová, Monika Pipová, Pavlina Jevinová and Jana Becová  which we would like to submit for consideration to the Foods in special issue "Detection, Control, Risk Assessment, and Prevention of Foodborne Microorganisms".

We would like to thank the reviewer for comments making this manuscript clearer and more reliable. All recommendations have been accepted by the authors. English language has been revised and errors have been corrected.  All edits are marked in the document via the „Track Changes“ function. Thank you for considering this manuscript.

Yours Sincerely,

Jana Výrostková, DVM, PhD.

Reviewer 2 Report

The manuscript entitled “Detection of resistant and enterotoxigenic strains of Staphylococcus warneri isolated in food of animal origin” is well designed, structured and written, by Regecová et al., appropriate English with clear structure. The authors investigated the antimicrobial susceptibility and enterotoxin production in Staphylococcus warneri isolated from food samples of animal origin. The results are interesting, Staphylococcus warneri  is a new emerging foodborne pathogen and it is so valuable to be investigated and characterized throughout the food samples, however, there are some major and minor points that need to be addressed to improve the quality of this valuable paper.

  • Major concern
  • My main concern in this manuscript is about the molecular characterization and development of multiplex PCR procedure for detection of enterotoxin genes of S. warneri as it is described in the materials and methods section previously developed by Sharma et al. You must use all positive controls of the reference bacteria harboured enterotoxins a, b, s, d, e in your methodology. Also, you should include processed and unprocessed gel electrophoresis blot images in your manuscript proving that you developed and used the procedure for gene detection correctly and successfully. Each enterotoxin gene should be determined and shown in the blot image with different nucleotide sizes. Also, you should provide and show the successful multiplex PCR products for reference strains, and negative and positive samples in your blot image. Please provide this issue completely in your manuscript.
  • Minor points
  • The abstract section should be restructured and rewritten. There are some unclear sentences, especially in the first part of this section. This section should not be started with the aim of this study.
  • In the introduction section: please describe more about the relationship between the S. warneri and foodborne diseases and the prevalence of this pathogen in different foods. What is the main source of contamination in the food chain?
  • In introduction section: please address a literature review regarding the association between this pathogen and foods. Why are foods important in the prevalence of foodborne diseases caused by this pathogen? Were there any outbreaks reported regarding this pathogen?
  • In materials and methods section: please address the DNA extraction in a separate part and describe this step in detail.

Regards,

Author Response

(The authors gave the same response as above.)

Round 2

Reviewer 1 Report

The manuscript was sufficiently improved and it can be considered suitable for publication in the present form.

Author Response

Jana Výrostková, DVM, PhD.

Department of Food Hygiene, Technology and Safety

University of Veterinary Medicine and Pharmacy

Komenského 73

041 81 Košice

Slovak Republic

E-mail: jana.vyrostkova@uvlf.sk

May 02, 2022

Editorial Board

Foods

MDPI

Postfach, CH-4020 Basel,

Switzerland

Dear Reviewer

Please find attached our revised research article Detection of resistant strains of Staphylococcus warneri isolated in food of animal origin “written by Ivana Regecová, Jana Výrostková, František Zigo, Gabika Gregová, Monika Pipová, Pavlina Jevinová and Jana Becová  which we would like to submit for consideration to the Foods in special issue "Detection, Control, Risk Assessment, and Prevention of Foodborne Microorganisms".

We would like to thank the academic editors for their comments, which make this manuscript clearer and more reliable. All recommendations were accepted by the authors. All edits are marked in the document using the "Track Changes" feature. However, line 129-220 belongs to the part of material and methodology, in our opinion this text describes the methodological procedure of phenotypic identification of staphylococci.

Thank you for dealing with this manuscript.

Yours Sincerely,

Jana Výrostková, DVM, PhD.

Reviewer 2 Report

All minor and major revisions have been addressed and done correctly. 

This manuscript can be accepted for publication in its present form.

Author Response

(The authors gave the same response as above.)
